# Frailty and Outcomes in Elderly ICU Patients: Insights from a Portuguese Cohort

**DOI:** 10.3390/healthcare13233063

**Published:** 2025-11-26

**Authors:** Eva Lourenço, Isabel Rodrigues, Mário Sampaio, Emília-Isabel Martins Teixeira-da-Costa

**Affiliations:** 1ICU Department, University Hospital Centre of the Algarve, 8000 Faro, Portugal; isrodrigues@ulsalg.min-saude.pt; 2Faculty of Medicine and Biomedical Sciences, Department of Medicine, University of Algarve, 8000 Faro, Portugal; 3Tavira Health Centre, Local Health Unit of the Algarve, 8800 Tavira, Portugal; mrsampaio@ulsalg.min-saude.pt; 4School of Health, Nursing Department, University of Algarve, 8000 Faro, Portugal; emiliaisabel.dacosta@denf.uhu.es; 5Department of Nursing, Faculty of Nursing, University of Huelva, 21004 Huelva, Spain; 6Health Sciences Research Unit: Nursing, Nursing School of Coimbra, 3001 Coimbra, Portugal

**Keywords:** frailty, elderly, ICU mortality, organ support, Clinical Frailty Scale, outcomes

## Abstract

**Highlights:**

**What are the main findings?**
Frailty was frequent (30.4%; 95% CI 23.0–38.9) among elderly ICU patients and was linked to higher comorbidity burden, greater illness severity, and increased need for invasive organ support.Mortality was substantially higher in frail patients, with ICU mortality reaching 50.0% (95% CI 34.6–65.4) and in-hospital mortality 76.3% (95% CI 60.8–87.0), compared with non-frail patients.

**What are the implications of the main finding?**
Routine frailty assessment at ICU admission is strongly associated with adverse outcomes at ICU admission and may support risk stratification.Incorporating frailty into ICU decision-making fosters patient-centered care and more appropriate allocation of intensive care resources.

**Abstract:**

**Background:** Frailty is a key determinant of outcomes in critically ill elderly patients, but data from Portugal remain limited. To our knowledge, this is the first study to examine the prevalence and prognostic impact of frailty among elderly ICU patients in a Portuguese hospital setting. **Objective:** To determine the prevalence of frailty among elderly patients admitted to an intensive care unit (ICU) in southern Portugal and to examine its crude associations with illness severity, organ support, and mortality outcomes. **Methods:** We conducted a retrospective cohort study including 125 patients aged ≥ 65 years admitted to the polyvalent ICU of Hospital de Faro over the last six months of 2024. Data included demographics, comorbidities, Charlson Comorbidity Index (CCI), severity scores (SOFA, SAPS II, APACHE II), and frailty status assessed by the Clinical Frailty Scale (CFS). Outcomes were the need for organ support, ICU and hospital mortality, and length of stay. **Results:** Frailty (CFS ≥ 5) was identified in 30.4% of patients. Frail patients were older, had higher comorbidity burden (CCI), and presented with significantly higher severity scores at admission. They also required more invasive support, including vasopressors and invasive mechanical ventilation, while acute kidney injury (AKI) requiring renal replacement therapy (RRT) was similar between groups. ICU mortality was significantly higher among frail patients (50.0% vs. 31.0%), as was hospital mortality (76.3% vs. 33.3%). Length of ICU stay did not differ, although frail patients tended to have longer hospitalizations overall. **Conclusions:** Frailty was highly prevalent and strongly associated with increased severity, greater need for organ support, and higher mortality. Routine frailty assessment at ICU admission may enhance prognostic accuracy and support patient-centered decision-making.

## 1. Introduction

With the rapid aging of populations worldwide, there is a pressing need to deepen our understanding of the clinical characteristics and outcomes of elderly patients admitted to intensive care units (ICUs). This demographic is uniquely vulnerable, not only because of the physiological changes associated with aging and the cumulative burden of comorbidities, but also due to the multidimensional condition of frailty, which has increasingly been recognized as central in critical care medicine [1,2]. Frailty represents a state of heightened vulnerability to external stressors and a reduced capacity for recovery, resulting from diminished physiological reserves across multiple systems. It has emerged as a pivotal determinant of outcomes in critically ill older adults, particularly regarding mortality and post-ICU recovery. Importantly, frailty is distinct from chronological age or the mere presence of comorbid conditions, positioning it as a multidimensional construct that better reflects patients’ physiological reserve and resilience to acute illness [3,4].

Recent epidemiological studies indicate that frailty, commonly assessed using the Clinical Frailty Scale (CFS) [4] and other validated instruments, is highly prevalent among older adults and serves as an independent predictor of both short- and long-term adverse outcomes in the ICU [5,6]. Although frailty affects roughly 25% of community-dwelling older individuals and nearly half of those aged over 85, its prevalence is markedly higher in critically ill populations, with European and North American studies reporting rates between 30% and 40% [6,7,8].

The clinical significance of frailty extends beyond risk stratification; it fundamentally shapes decision-making, allocation of intensive care resources, and communication with patients and families regarding prognosis and goals of care [7,9,10]. Recent multicenter research, including the VIP1 and VIP2 studies, has consolidated frailty’s role as an independent predictor of mortality, organ support requirements, and functional decline after ICU admission [9,11]. Of particular concern, frail patients are more likely not only to succumb during hospitalization but also to experience long-term disability, raising critical questions about the appropriateness of intensive and potentially burdensome interventions [12].

Despite this strong international evidence, epidemiological data on ICU frailty in Portugal remain scarce. In Portugal, the resident population was estimated at 10.7 million in 2024, and individuals aged 65 years or older represented 24.1% of the total population, the second highest proportion in the European Union, with an aging index of 188.1 [13]. This demographic scenario highlights Portugal as one of the most aged countries in Europe, with direct implications for healthcare organization and the growing number of older adults admitted to intensive care units. However, as emphasized above, frailty extends beyond chronological aging: it reflects a multidimensional and functional vulnerability that more accurately predicts adverse outcomes in critical illness. Understanding its prevalence and clinical consequences in the Portuguese ICU setting is therefore essential to improving the appropriateness and quality of care for this growing patient group. To our knowledge, this is the first study to describe the prevalence and impact of frailty among elderly ICU patients in a Portuguese hospital setting. The aim of this study was to determine the prevalence of frailty, defined as a Clinical Frailty Scale (CFS) score ≥ 5, among elderly ICU patients in a major hospital in southern Portugal, and to examine its crude associations with illness severity (SOFA, SAPS II, APACHE II), need for organ support, and ICU and hospital mortality using standard bivariate statistical tests.

## 2. Materials and Methods

This retrospective cohort study included all consecutive patients aged 65 years or older who were admitted to the polyvalent ICU of Hospital de Faro (Unidade Local de Saúde Algarve, Faro, Portugal) during the last six months of 2024. All patients meeting the inclusion criteria within this period were included without sampling or exclusion beyond predefined criteria. Eligible patients were required to have remained in the ICU for at least 24 h and to have complete data available in electronic medical records regarding demographic characteristics, comorbidities, illness severity scores, and frailty assessment. Patients younger than 65 years, those admitted for less than 24 h (e.g., short postoperative observation), and those with insufficient data to determine frailty status or to calculate severity scores were excluded.

Data were extracted from electronic medical records using a standardized protocol. Collected variables included sociodemographic information (age, sex), comorbidities (assessed with the Charlson Comorbidity Index [14] and detailed secondary diagnoses), type and urgency of admission (medical vs. surgical, trauma, and specific causes of admission), and acute events such as sepsis, cardiac arrest, acute kidney injury, and need for organ support. Illness severity at admission was assessed using the Simplified Acute Physiology Score II (SAPS II), the Sequential Organ Failure Assessment (SOFA), and the Acute Physiology and Chronic Health Evaluation II (APACHE II) [15,16,17]. Frailty was measured with the nine-point Clinical Frailty Scale (CFS), a validated and widely used instrument [4]. The CFS score was assigned by the attending ICU physicians at the time of admission, based on patients’ pre-admission functional status. Information was obtained from electronic medical records, including prior clinical documentation, nursing notes, and, when available, reports from family members or caregivers. This approach follows the recommendations [4] and reflects standard practice in retrospective assessments of frailty in critical care settings. In line with established guidelines, patients with a CFS score of 5 or higher were categorized as frail, while those with scores between 1 and 4 were classified as non-frail [4].

Outcomes of interest included the need for vasopressors, invasive or non-invasive mechanical ventilation, duration of ventilation, use of renal replacement therapy, length of ICU and hospital stay, and mortality during ICU and hospital admission. Secondary outcomes included the incidence of nosocomial infections.

Nosocomial infections were defined according to CDC criteria [18]. Acute kidney injury (AKI) was defined based on KDIGO guidelines as an increase in serum creatinine ≥ 0.3 mg/dL within 48 h, or ≥1.5 times baseline within 7 days, or urine output < 0.5 mL/kg/h for 6 h [19]. Renal replacement therapy (RRT) was recorded when any form of extracorporeal support was initiated for AKI management [19].

Descriptive statistics (means, standard deviations, medians, and frequency distributions) were used to characterize the study population. Comparisons between frail (CFS ≥ 5) and non-frail (CFS 1–4) groups were conducted using Student’s *t*-test for continuous variables and the Pearson Chi-square test for categorical variables. Missing data were minimal (<5%). Patients with incomplete information on key variables were excluded from the analysis (complete-case analysis); no data imputation procedures were applied.

All analyses were exploratory and unadjusted. Given the sample size and the descriptive nature of the study, no correction for multiple comparisons was applied.

Given the retrospective, single-center design and the limited six-month inclusion period, the results should be interpreted with caution regarding external validity. All statistical analyses were performed using IBM SPSS Statistics, version 29.0.1.0 (171) (IBM Corp., Armonk, NY, USA). A two-tailed *p*-value < 0.05 was considered statistically significant. Associations between frailty status and clinical outcomes were examined using unadjusted analyses. Multivariable models were not performed due to the limited sample size and the relatively small number of outcome events, which could compromise model stability and validity.

The study was approved by the Ethics Committee of the Algarve Local Health Unit (ULSALG) (approval no. 075/2025), with a favorable opinion issued on 9 June 2025. A formal request for waiver of informed consent was submitted and accepted, as the study relied exclusively on secondary clinical data that were fully anonymized. No direct or indirect identifiers were collected or processed, making it impossible to trace data back to individual patients. Although the dataset referred to patients admitted during the last semester of 2024, retrieval and analysis were initiated only after the ethics approval had been granted. All procedures complied with the principles of the Declaration of Helsinki, the EU General Data Protection Regulation, and the applicable national legislation.

## 3. Results

A total of 125 patients aged 65 years or older met the inclusion criteria and were analyzed. The results are presented in four parts: first, the baseline demographic and clinical characteristics of the cohort; second, the reasons for ICU admission and severity scores at presentation; third, the prevalence and distribution of frailty according to the Clinical Frailty Scale (CFS); and finally, the comparison of clinical characteristics and outcomes between frail and non-frail patients, with emphasis on severity, organ support, and mortality.

### 3.1. Baseline Characteristics

Our cohort consisted predominantly of men (58.4%), with a mean age of 75.2 years (SD 6.5; range 65–92), and nearly one in four patients aged 80 years or older, underscoring the advanced age of the study population. The overall burden of comorbidities was high, reflecting the typical multimorbidity profile of critically ill elderly patients and aligning with international data. Hypertension, diabetes, heart failure, arrhythmia, chronic kidney disease, chronic lung disease, and malignancy were among the most common secondary diagnoses. Specifically, hypertension was documented in 42.4% of patients, diabetes in 11.2%, and chronic heart failure in 4.8%. These prevalences are parallel to those reported in large European cohorts, such as the ICON [20] and VIP2 studies [7,9] (Table 1).

### 3.2. ICU Admission Characteristics

The main reasons for ICU admission in this cohort were postoperative monitoring (39.2%) and acute respiratory failure (18.4%), followed by septic shock (13.6%), cardiac arrest (8.8%), trauma (6.4%), and acute kidney injury (AKI) requiring renal replacement therapy (RRT) (6.4%). Nosocomial infections occurred in approximately one in three patients (34.4%, 43/125), most commonly respiratory (18.4%) and urinary tract infections (10.4%). At admission, illness severity was high, with mean SOFA, SAPS II, and APACHE II scores of 6.75, 47.9, and 18.9, respectively. Detailed admission characteristics are presented in Table 2.

### 3.3. Frailty Prevalence and Distribution

Frailty, as assessed by the Clinical Frailty Scale (CFS), was present in 30.4% of patients in the cohort (CFS ≥ 5). The distribution across the nine CFS categories closely mirrored that observed in large multicenter cohorts: nearly half of the patients were robust (8.8% very fit and 38.4% well), 19.2% were managing well (CFS 3), 3.2% were vulnerable (CFS 4), 12.8% mildly frail, 16.0% moderately frail, and only a minority were severely (0.8%) or very severely frail (0.8%). No patient was categorized as terminally ill (CFS 9).

The full distribution of CFS categories (1–9) is illustrated in Figure 1, providing a visual overview of the cohort’s frailty profile, while detailed frequencies are reported in Table 3.

Severity at admission was high, with mean SOFA, SAPS II, and APACHE II scores of 6.75, 47.9, and 18.9, respectively. Taken together, these findings underscore that frailty was both frequent and clinically relevant in this cohort.

### 3.4. Outcomes

At admission, frail patients presented with a significantly higher burden of comorbidities compared with non-frail patients, as reflected by the Charlson Comorbidity Index (5.0 ± 1.6 vs. 3.9 ± 1.2; t = −4.14, *p* < 0.001). Severity scores were also consistently higher among frail patients, including SOFA (8.7 ± 4.1 vs. 5.9 ± 4.3; t = −3.47, *p* = 0.001), SAPS II (57.3 ± 18.2 vs. 43.8 ± 22.5; t = −3.29, *p* = 0.001), and APACHE II (21.8 ± 8.1 vs. 17.6 ± 11.3; t = −2.06, *p* = 0.041).

Frail patients (CFS ≥ 5) experienced significantly higher mortality compared with their non-frail counterparts, with ICU mortality reaching 50.0% (19/38) versus 31.0% (27/87) (χ^2^ = 4.09, *p* = 0.044; OR = 2.22, 95% CI 1.02–4.85). In-hospital mortality showed an even greater disparity, rising to 76.3% (29/38) among frail patients compared with 33.3% (29/87) in the non-frail group (χ^2^ = 19.6, *p* < 0.001; OR = 6.45, 95% CI 2.69–15.38).

The need for invasive organ support was also higher among frail patients, including vasopressor use (65.8% vs. 37.9%; χ^2^ = 8.25, *p* = 0.004; OR = 3.14, 95% CI 1.42–7.00) and invasive mechanical ventilation (89.5% vs. 58.6%; χ^2^ = 11.57, *p* = 0.001; OR = 5.99, 95% CI 1.96–18.52). The duration of invasive ventilation was comparable between groups. The median [IQR] duration was 4 [2–12] days in non-frail patients and 4 [2–8.25] days in frail patients (Mann–Whitney U = 800.0, Z = −0.61, *p* = 0.545), indicating no statistically significant difference in ventilation time between the two groups. By contrast, the requirement for renal replacement therapy did not differ between groups (10.5% vs. 10.3%; χ^2^ = 0.001, *p* = 0.976; OR = 0.98, 95% CI 0.28–3.41).

Length of ICU stay was comparable between groups (7.9 ± 9.9 vs. 6.8 ± 8.9 days; t = −0.58, *p* = 0.563), while frail patients tended to have longer overall hospitalizations (36.6 ± 54.1 vs. 23.2 ± 25.8 days; t = −1.45, *p* = 0.153), although this difference did not reach statistical significance (Table 4).

## 4. Discussion

Our study confirms frailty, measured by the Clinical Frailty Scale (CFS), as a highly prevalent and prognostically relevant condition among elderly ICU patients in Portugal, with a prevalence of 30.4%. This figure falls within the range reported in other European multicenter studies, where frailty rates vary between 24% and 43% [6,21], supporting the comparability of our findings and reinforcing that frailty is a common condition in critically ill older adults.

The clinical consequences of frailty in our cohort were striking. Frail patients were older, presented with higher illness severity at admission, and required more invasive organ support, including vasopressors and mechanical ventilation. These findings mirror results from the VIP and ICON studies and emphasize frailty as a marker of both acute illness complexity and diminished physiological reserve [7,9,20]. These findings reaffirm the prognostic importance of frailty in the critical care context

Although frail patients required invasive mechanical ventilation more frequently, the duration of ventilation did not differ significantly between groups. The median [IQR] duration was 4 [2–12] days in non-frail patients (range 1–37) and 4 [2–8.25] days in frail patients (range 1–44) (Mann–Whitney U = 800.0, Z = −0.61, *p* = 0.55). This finding suggests that, once invasive support is initiated, the course of respiratory management tends to be similar regardless of frailty status. Comparable ventilation times have also been reported in other ICU cohorts, possibly reflecting standardized weaning protocols and clinical decision-making processes that are not primarily driven by frailty scores. Nevertheless, the higher frequency of invasive ventilation among frail patients underscores their increased vulnerability and greater overall burden of critical illness, which may still contribute indirectly to poorer outcomes. Mortality outcomes were also strongly associated with frailty. ICU mortality reached 50% among frail patients compared with 31% in non-frail patients, while in-hospital mortality rose to 76.3% versus 33.3%. These results are somewhat higher than those observed in the VIP2 study (ICU mortality 40%; 30-day mortality 46%) but remain broadly consistent with international patterns. Such differences may reflect local case-mix, comorbidity profiles, and admission thresholds. The predominance of cardiovascular and metabolic comorbidities in our cohort, similar to the epidemiological profile of Mediterranean countries, may also have contributed to the slightly higher mortality rates observed [7,9,22].

In line with previous systematic reviews and meta-analyses, our findings confirm that frailty is strongly associated with higher mortality among elderly ICU patients. Frail patients also experienced longer and more complex hospital trajectories, consistent with previous evidence showing that frailty is linked to prolonged recovery, functional decline, and long-term disability after critical illness.

In our cohort, ICU length of stay was similar between groups, but frail patients showed a tendency toward longer overall hospitalizations [1,23].

Several pathophysiological mechanisms may explain these associations. Frail individuals exhibit reduced physiological reserves, impaired stress responses, and often subclinical organ dysfunction, all of which compromise resilience to critical illness. In addition, frailty is closely interconnected with malnutrition and metabolic dysregulation. Loss of muscle mass, nutrient deficiencies, systemic inflammation, and impaired protein synthesis further weaken physiological resilience and contribute to adverse outcomes. This metabolic–nutritional dimension of frailty has been increasingly recognized in critical care literature and aligns with the ESPEN 2019 and 2023 guidelines, which emphasize early nutritional optimization and personalized metabolic support in the ICU [24,25,26]. These biological vulnerabilities help explain why frail patients are more likely to require invasive support and why such interventions may not always translate into improved survival or functional recovery. Our data therefore add to growing evidence that early, structured discussions with patients and families about prognosis, treatment intensity, and quality of life are crucial in the ICU context [1,7].

From a clinical standpoint, systematic frailty assessment at ICU admission can serve as a cornerstone for individualized and ethically sound care. Identifying frail patients early supports proportional decision-making regarding the intensity of life-sustaining therapies, encourages timely multidisciplinary discussions, and fosters transparent communication with families about prognosis and goals of care. In practice, this process can be conceptualized as a structured pathway: CFS screening → multidisciplinary discussion → alignment of treatment goals → personalized care plan. Implementing such an approach may enhance both clinical outcomes and the quality of decision-making in intensive care.

Between-country differences in ICU admission practices have also been described, even within Europe, influenced by local triage protocols, pre-admission functional status, and cultural or societal values regarding end-of-life care [7,9]. The heterogeneity of frailty levels observed in our sample, from robust elderly individuals to those severely frail, underscores the need for future studies to clarify which subgroups of frail patients may still benefit from intensive care and which may be better managed through alternative or palliative strategies.

From a practical perspective, routine frailty screening at ICU admission could be easily implemented using the Clinical Frailty Scale (CFS), which requires minimal time and no additional resources. Integrating the CFS into the initial clinical assessment may complement traditional severity scores (such as SOFA, SAPS II, and APACHE II) and facilitate early multidisciplinary decision-making. This approach can support timely discussions regarding treatment intensity, rehabilitation potential, and nutritional or palliative interventions, ultimately contributing to more patient-centered and goal-concordant care.

Given the sample size and the limited number of outcome events, all analyses were unadjusted. Multivariable modeling was not performed to avoid overfitting and unstable estimates, and this should be considered when interpreting the observed associations.

This study has several limitations. Its retrospective, single-center design limits the generalizability of findings, and the six-month inclusion period may not capture seasonal or annual variation in ICU admissions. Although the sample size is comparable to other single-center cohorts, it restricted the possibility of performing additional subgroup analyses and reduced statistical power for less frequent outcomes. In addition, reliance on electronic health records may have introduced information bias due to incomplete or inconsistent documentation. Nutritional parameters, such as albumin levels or caloric intake during the first 48 h, were not systematically recorded in medical charts and were therefore not included in the analysis. The absence of standardized nutritional data represents a potential confounding factor in the observed associations between frailty and clinical outcomes. Nonetheless, the richness of our dataset and the strong alignment of our results with international literature support the robustness of our findings.

## 5. Conclusions

Frailty, as measured by the Clinical Frailty Scale, was highly prevalent in this Portuguese ICU cohort, affecting nearly one-third of elderly patients. It was consistently associated with greater illness severity, increased need for invasive organ support, and substantially higher ICU and in-hospital mortality. While the duration of invasive ventilation was similar between groups, frail patients experienced a higher overall burden of organ support and worse outcomes. Although ICU length of stay did not differ significantly, frail patients showed a tendency toward longer hospitalizations, reflecting more complex trajectories of care. Although no adjusted analysis was performed, frailty demonstrated a consistent and clinically meaningful association with adverse outcomes, supporting its potential prognostic value in critically ill elderly patients. These findings reinforce the importance of incorporating systematic frailty assessment at ICU admission. Doing so not only improves prognostic accuracy but also supports patient-centered decision-making, early discussions about goals of care, and alignment of treatment intensity with individual values and preferences. At the health system level, such assessments can contribute to more equitable and appropriate allocation of intensive care resources.

Further multicenter studies in Portugal and comparable Mediterranean settings are warranted to confirm these results, refine predictive models that combine frailty with severity-of-illness scores, and explore which subgroups of frail patients may benefit most from intensive care. Such evidence will be critical to guide clinical practice, optimize patient outcomes, and balance intensive interventions with alternative or palliative approaches where appropriate.

## Figures and Tables

**Figure 1 healthcare-13-03063-f001:**
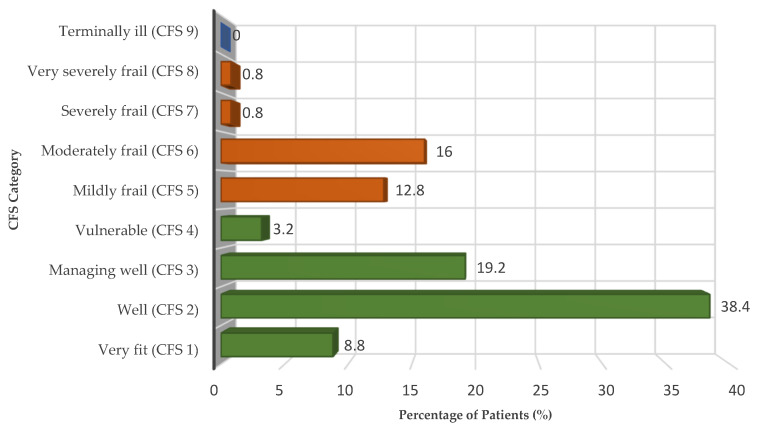
Distribution of elderly ICU patients across Clinical Frailty Scale (CFS) categories (*n* = 125 Orange bars correspond to frail patients (CFS ≥ 5), whereas green bars represent non-frail patients (CFS ≤ 4). Values are expressed as percentages of the total sample.

**Table 1 healthcare-13-03063-t001:** Baseline demographic and clinical characteristics of the study cohort (*n* = 125).

Variable	*n* (%)/Mean ± SD	Range
**Demographics**		
Age (years)	75.2 ± 6.5	65–92
Age ≥ 80 years	31 (24.8%)	-
Male sex	73 (58.4%)	-
**Comorbidities**		
Hypertension	53 (42.4%)	-
Diabetes mellitus	14 (11.2%)	-
Chronic heart failure	6 (4.8%)	-
Arrhythmia	3 (2.4%)	-
Chronic kidney disease	8 (6.4%)	-
Chronic lung disease	6 (4.8%)	-
Malignancy	14 (11.2%)	-
Cerebrovascular disease	4 (3.2%)	-
Psychiatric disorders	3 (2.4%)	-
Other comorbidities	9 (7.2%)	-
No comorbidities identified	5 (4.0%)	-
Charlson Comorbidity Index	4.26 ± 1.39	2–10

Data are presented as mean ± standard deviation (SD) or number (percentage).

**Table 2 healthcare-13-03063-t002:** Admission diagnoses, infections, and severity scores of elderly ICU patients (*n* = 125).

Primary Reason for ICU Admission	*n* (%)/Mean ± SD
Postoperative monitoring	49 (39.2%)
Acute respiratory failure	23 (18.4%)
Septic shock (respiratory/abdominal)	17 (13.6%)
Cardiac arrest	11 (8.8%)
Trauma	8 (6.4%)
AKI requiring RRT	8 (6.4%)
Other medical emergencies	9 (7.2%)
**Infections during ICU stay**	
Nosocomial respiratory infections	23 (18.4%)
Urinary tract infections	13 (10.4%)
Abdominal/other infections	7 (5.6%)
**Severity scores at admission**	
SOFA score	6.75 ± 4.41
SAPS II	47.9 ± 22.1
APACHE II	18.9 ± 10.6

Data are presented as mean ± standard deviation (SD) or number (percentage). ICU = Intensive Care Unit; SOFA = Sequential Organ Failure Assessment; SAPS II = Simplified Acute Physiology Score II; APACHE II = Acute Physiology and Chronic Health Evaluation II.

**Table 3 healthcare-13-03063-t003:** Distribution of elderly ICU patients across Clinical Frailty Scale (CFS) categories (*n* = 125). The same data are illustrated in Figure 1.

CFS Category	*n* (%)
Very fit (CFS 1)	11 (8.8%)
Well (CFS 2)	48 (38.4)
Managing well (CFS 3)	24 (19.2%)
Vulnerable (CFS 4)	4 (3.2%)
Mildly frail (CFS 5)	16 (12.8%)
Moderately frail (CFS 6)	20 (16.0%)
Severely frail (CFS 7)	1 (0.8%)
Very severely frail (CFS 8)	1 (0.8%)
Terminally ill (CFS 9)	0 (0%)

The Clinical Frailty Scale (CFS) ranges from 1 (very fit) to 9 (terminally ill), providing a global measure of patients’ frailty status based on comorbidity, function, and cognition. For the purposes of this study, patients with CFS ≥ 5 were categorized as frail.

**Table 4 healthcare-13-03063-t004:** Comparison of clinical severity, comorbidity burden, and outcomes according to frailty status (CFS ≥ 5) in critically ill elderly patients (*n* = 125).

Variable	Non-Frail (*n* = 87)	Frail (*n* = 38)	Test Statistic	*p*-Value	OR (95% CI)
**Severity scores at admission**					
SOFA score (mean ± SD)	5.89 ± 4.29	8.74 ± 4.07	*t* = −3.471	0.001	-
SAPS II (mean ± SD)	43.8 ± 22.5	57.3 ± 18.2	*t =* −3.285	0.001	-
APACHE II (mean ± SD)	17.6 ± 11.3	21.8 ± 8.1	*t =* −2.061	0.041	-
Charlson Comorbidity Index (mean ± SD)	3.94 ± 1.19	5.00 ± 1.58	*t* = −4.14	<0.001	-
**Organ support**					
Vasopressor use (*n*/%)	(33) 37.9% (95% CI 28.5–48.4)	(25) 65.8% (95% CI 49.9–78.8)	χ^2^ = 8.25	0.004	3.14 (1.42–7.00)
Invasive mechanical ventilation (*n*/%)	(51) 58.6% (95% CI 48.1–68.4)	(34) 89.5% (95%CI 75.9–95.8)	χ^2^ = 11.57	0.001	5.99 (1.96–18.52)
Duration of invasive ventilation (days, median [IQR])	4 [2–12]	4 [2–8.25]	U = 800.0	0.545	-
Renal replacement therapy (*n*/%)	(9) 10.3% (95% CI 5.5–18.5)	(4) 10.5% (95% CI 4.2–24.1)	χ^2^ = 0.001	0.976	0.98 (0.28–3.41)
**Outcomes**					
ICU mortality (*n*/%)	(27) 31% (95% CI 22.3–41.4)	(19) 50.0% (95% CI 34.8–65.2)	χ^2^ = 4.09	0.044	2.22 (1.02–4.85)
Hospital mortality (*n*/%)	(29) 33.3% (95% CI 24.3–43.8)	(29) 76.3% (95% CI 60.8–87.0)	χ^2^ = 19.6	<0.001	6.45 (2.69–15.38)
ICU length of stay (days, mean ± SD)	6.83 ± 8.94	7.87 ± 9.85	t = −0.58	0.563	-
Hospital length of stay (days, mean ± SD)	23.20 ± 25.77	36.55 ± 54.07	t = −1.45	0.153	-

Data are presented as mean ± standard deviation (SD) or number (percentage). Comparisons were performed using Student’s *t*-test for continuous variables and Pearson’s Chi-square test (χ^2^) or Fisher’s exact test for categorical variables, as appropriate. Crude odds ratios (OR) with 95% confidence intervals (CIs) were calculated for dichotomous outcomes only. ICU = Intensive Care Unit; CFS = Clinical Frailty Scale; SOFA = Sequential Organ Failure Assessment; SAPS II = Simplified Acute Physiology Score II; APACHE II = Acute Physiology and Chronic Health Evaluation II.

## Data Availability

The data presented in this study are not publicly available due to privacy and ethical restrictions, as they consist of anonymized clinical records from the Algarve Local Health Unit (ULSALG). Data may be made available upon reasonable request from the corresponding author and with prior approval of the ULSALG Ethics Committee.

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
