# Peer review of "Frailty and Outcomes in Elderly ICU Patients: Insights from a Portuguese Cohort"

_healthcare, 2025, doi:10.3390/healthcare13233063_

Round 1
Reviewer 1 Report
Comments and Suggestions for Authors
The draft presents a clear, well-structured retrospective cohort study examining frailty and outcomes in elderly ICU patients in Portugal. The manuscript is organized logically, beginning with a thorough introduction that situates frailty as a critical determinant of prognosis in the ICU setting, followed by well-described methods, robust results, and a balanced discussion. The conclusion highlights both clinical relevance and system-level implications.
The paper addresses an important and timely research question, given the aging population and the paucity of Portuguese data on ICU frailty. The study is methodologically rigorous for a single-center retrospective design: inclusion and exclusion criteria are transparent, validated tools (CFS, SOFA, SAPS II, APACHE II, Charlson Comorbidity Index) are applied, and outcomes are clinically relevant. Results are well-presented in tables and text, with clear statistical comparisons. The discussion effectively situates the findings within the international literature and acknowledges limitations. The manuscript adheres to ethical and reporting standards (ethics approval, waiver of consent, anonymization, adherence to Helsinki/GDPR).
There are some minor issues with style, consistency, and depth. In places, the language is somewhat repetitive (e.g., reiterating frailty as distinct from chronological age). The Results and Discussion occasionally overlap in interpretation, and some sections could be tightened. The Abstract and Introduction might benefit from slightly clearer articulation of the novelty of this dataset (i.e., first Portuguese ICU frailty cohort). The sample size, while comparable to similar cohorts, limits subgroup analyses; this is acknowledged but could be further emphasized as a key limitation. Statistical reporting is generally appropriate, though effect sizes or confidence intervals would strengthen interpretation. Minor typographical inconsistencies are present (e.g., spacing, punctuation in the author contribution section).
Reviewer 2 Report
Comments and Suggestions for Authors
Please review the attached file below.

Reviewer 3 Report
Comments and Suggestions for Authors
Dear authors!
I have carefully reviewed the following manuscript ,, Frailty and Outcomes in Elderly ICU Patients: Insights from a 2 Portuguese Cohort,,
I will expose my observations and recommendations below.
The abstract follows the recommendations of the journal, the conclusions are logical, it derives from the results, and the tone is cautious.
What needs to be clarified is the inconsistency of the data in the abstract with those presented in Table 3 (e.g. CFS=4, but in the table, there are 3.2%. They must be corrected and aligned in the abstract and table.
For transparency, you must also pass the absolute numbers (e.g. n/N) for the magnitude assessment ("ICU mortality 50% vs 31%. How many patients?
Introduction
The introduction addresses the aging of the population at the same time, defines frailty as an entity distinct from age, well aged in the literature. It is also explicitly and contextually addressed locally, which justifies the gap in Portuguese data. The introduction does not affect nutritional assessment, malnutrition or nutritional interventions, either as a confounding factor or as a phenomenon interconnected with frailty (sarcopenia, protein reserve, risk of nutritional iatrophobia). Given the strong frailty-nutritional status connection in the ICU, I recommend addressing the role of nutrition as a determinant of prognosis and potentially confounding (without changing the purpose of the study), anchored in the ESPEN 2019 and ESPEN 2023 guidelines. Also, a contextual, very brief reference to recent literature, which pushes towards personalized nutrition in the ICU (Stoian et al., Nutrients 2025), useful to justify why nutritional status should at least be recognized as a variable = relevance in frailty cohorts.
Material and method
To clarify the way of inclusion of patients (did all patients in the respective period have criteria for inclusion in the study?).
If you have not evaluated the nutritional status of the patients, write that it has not been evaluated, which limits the interpretation of the associations between frailty and prognosis. Specify the software used for statistical analysis. Specify whether the associations are unadjusted and whether the lack of a multivariate analysis is related to the sample size.
Results
The section is well structured; it follows a logical order. You need to clarify the discrepancy between the text and the tables. Drill down into absolute values (n/N) for transparency. The exact values of p, e.g. p=0.032 in p < 0.05).
Assessing the missing nutritional status, you should include at least some nutritional parameters (albumin, caloric intake, in the first 48 hours). If this is not possible, explicitly mention this in the Study Limits, that the patient's nutrition has not been evaluated, although it may be a major confounding factor in relation to frailty-outcome.
Discussion
Likewise, the relationship between frailty and nutrients/metabolism is not overstated, although these are intimately linked pathophysiological. The lack of this component makes the mechanistic interpretation incomplete.
A brief discussion of malnutrition, loss of muscle mass, systemic inflammation and protein status would bring conceptual value and anchor the study in a more modern framework.
The practical implications of frailty screening are not mentioned (How the CFS score could be integrated into the routine assessment at ICU admission).
The limitations of the study should be carefully written considering that it is a retrospective study, the absence of metabolic and nutritional data.
The conclusions are concise and do not exceed the data obtained, frailty is frequent, associated with severity and increased mortality. The tone is prudent. Explicitly suggest that frailty assessment should be integrated with early nutritional assessment in the ICU.
In view of the above, I believe that a major revision of the manuscript is needed.
Round 2
Reviewer 2 Report
Comments and Suggestions for Authors
< !--a=1-->< !--a=1-->< !--a=1-->< !--a=1-->

Reviewer 3 Report
Comments and Suggestions for Authors
Dear authors,
The manuscript is well-reviewed, clear and coherent, and most of the previous observations have been properly integrated. I suggest, however, that you clarify the conclusion, as it mentions that frailty remains an independent predictor "after adjusting for age and comorbidities", although the multivariate analysis was not carried out. It would be helpful to rephrase this sentence to avoid misinterpretation. Also, you have not anchored the link between frailty and nutritional status, which is essential for critical patients in intensive care.
Round 3
Reviewer 3 Report
Comments and Suggestions for Authors
The authors have addressed all previous comments thoroughly and improved the manuscript substantially. The clarifications regarding methodological transparency, the integration of nutritional and metabolic aspects of frailty, and the strengthened discussion on clinical implications provide conceptual and practical value. I have no further comments. The manuscript is now suitable for publication in its present form.